# Serological Examinations of Significant Viral Infections in Domestic Donkeys at the Special Nature Reserve “Zasavica”, Serbia

**DOI:** 10.3390/ani13132056

**Published:** 2023-06-21

**Authors:** Sava Lazić, Sara Savić, Tamaš Petrović, Gospava Lazić, Marina Žekić, Darko Drobnjak, Diana Lupulović

**Affiliations:** 1Scientific Veterinary Institute “Novi Sad”, Rumenački put 20, 21000 Novi Sad, Serbia; lazic@niv.ns.ac.rs (S.L.); sara@niv.ns.ac.rs (S.S.); tomy@niv.ns.ac.rs (T.P.); goga@niv.ns.ac.rs (G.L.); marina@niv.ns.ac.rs (M.Ž.); 2Center for Preservation of Indigenous Breeds—CEPIB, Vere Dimitrijević, 11186 Zemun, Serbia; drobnjakvet@yahoo.com

**Keywords:** domestic donkey, equine infectious anemia, African horse sickness, equine herpesvirus type 1 infection, equine influenza, equine viral arteritis, Serbia

## Abstract

**Simple Summary:**

The domestic donkey is an endangered equid species. There are 199 donkeys in the Special Nature Reserve “Zasavica”, out of the estimated 1000 that are assumed to exist in Serbia. The aim of this study was to assess the exposure of donkeys in the Special Nature Reserve “Zasavica” to some of the most significant viral pathogens and establish the seroprevalence of equine infectious anemia, African horse sickness, equine herpesvirus type 1 infection, equine influenza and equine viral arteritis. In tested blood samples, antibodies against equine herpesvirus type 1 and equine influenza virus (subtype H3N8) were found, while all samples tested negative for African horse sickness virus, equine infectious anemia virus and equine arteritis virus. To our knowledge, this study represents the first results of detection of these pathogens in the population of domestic donkeys in Serbia.

**Abstract:**

The paper presents the findings of specific antibodies in the blood sera of donkeys against the following viruses: equine infectious anemia virus (EIAV), African horse sickness virus (AHSV), equine herpesvirus type 1 (EHV-1), equine influenza virus subtype H3N8 (EIV) and equine arteritis virus (EAV). The analyses were conducted during the year 2022. From a total of 199 donkeys bred in “Zasavica”, blood was sampled from 53 animals (2 male donkeys and 51 female donkeys), aged 3 to 10 years. Specific antibodies against EIAV were not detected in any of the tested animals using the agar-gel immunodiffusion (AGID) assay. No specific antibodies against AHSV, tested by enzyme-linked immunosorbent assay (ELISA), or antibodies against EAV, tested by the virus neutralization test (VNT) and ELISA were detected in any of these animals. A positive serological result for EHV-1 was determined by the VNT in all animals, with antibody titer values ranging from 1:2 to 1:128, while a very low antibody titer value for EIV (subtype H3N8) of 1:16 was determined in 18 donkeys using the hemagglutination inhibition test (HI test).

## 1. Introduction

The modern donkey descended from the African wild donkey (*Equus asinus africanus*) around 4.5 million years ago. However, it was domesticated only 5000 years ago [1]. The donkey’s closest “relative” is the horse. The horse (*Equus caballus*) and the donkey (*Equus asinus*) both belong to the Equidae family. Although they share a great number of characteristics, there are also significant physiological and genetic differences. Horses and donkeys have different number of chromosomes: the horse has 64 chromosomes, while the donkey has 62). There are also differences in pulse, respiration and body temperature. Although they can suffer from the same diseases, thanks to the fact that donkeys originate from Sub-Saharan Africa, they are considered extremely hardy animals, adapted to poor living and nutritional conditions. As a result, donkeys are resistant to many diseases or the disease occurs either in a milder form, or even as asymptomatic. However, donkeys which are not used for physical work are more prone to hyperlipidemia and laminitis [2].

Donkeys have been living with humans for centuries and are used for hard physical work [3]. These working equids still play a significant role in the lives of around 600 million people in low-income countries across the world. In recent years, there has been a noticeable trend of keeping donkeys as pets. The demand for donkey milk and dairy products, as well as donkey milk cosmetics, has also increased. In some parts of the world, donkey meat is consumed. Donkeys are used in tourism and in animal therapy of children with developmental disabilities [4,5].

The number of animals in the equine population cannot be determined with certainty, but it is estimated that there are about 60.2 million horses, 53 million donkeys and 7.8 million mules worldwide [6]. A slight increase in the number of donkeys, of approximately 1% per year, has been observed in the period from 1997 to 2018 in the world. However, the largest increase in the number of donkeys was recorded in poorly developed, rural areas of Asia and Africa, while their number is declining in Eastern Europe. The most remarkable decline in the number of donkeys was recorded in Bulgaria and Greece [5]. According to the latest data from 2018, 4.813 donkeys and mules were registered in Serbia [7]. The exact number of donkeys in Serbia is not known, but according to the unofficial data, there are only about 1000 of them [8]. Out of this number, 199 animals, mainly of the Balkan breed, live in “Zasavica”, which comprises the largest population of donkeys housed in one location (Figure 1).

“Zasavica” is located in the Republic of Serbia, in the region of southern Bačka and Srem, on the territory of two municipalities, Sremska Mitrovica and Bogatić. The total area of the reserve is 3462.65 hectares with the river Zasavica flowing through the reserve. “Zasavica” is a member of the EUROPARC Federation (professional network of European Protected Areas). On a pasture called “Valjevac”, within an area of about 300 hectares, in a free-ranging regime 32 Podolian cattle, 199 domestic donkeys, 206 horses (mountain breed) and 38 Mangalitsa pigs have been bred. The objective of this study is to test a population of donkeys in “Zasavica” to determine if they have been exposed to five viruses: equine infectious anemia virus, African horse sickness virus, equine herpesvirus type 1, equine influenza virus (subtype H3N8) and equine arteritis virus. These studies, to our knowledge, are the first results of seroprevalence and detection of these pathogens in a limited population of domestic donkeys that have been conducted in Serbia.

### 1.1. Equine Infectious Anemia (EIA)

Equine infectious anemia (EIA) occurs in equids, such as horses, mules, donkeys and zebras, and is extremely contagious. Equine infectious anemia is an infectious multisystemic disease, listed as one of the 15 notifiable diseases of equids by the WOAH Manual of Diagnostic Test and Vaccines for Terrestrial Animals [9]. The causative agent of EIA is equine infectious anemia virus (EIAV), belonging to the *Retroviridae* family, genus *Lentivirus*. Equine infectious anemia can occasionally be found in horses in Serbia; however, there are no comprehensive studies on donkeys in our country [10]. Equine infectious anemia can only survive in the bodies of equids [11]. Although this disease represents a health and economic problem in equids, the occurrence of EIA in donkeys is still neglected. Many factors can affect the frequency of disease occurrence, such as the presence of the virus in the region, the density of the horse population, the presence of vectors and disease control and monitoring programs [12]. The primary route of transmission of the virus, i.e., natural infection, is considered to be through the blood of infected animals with the help of the disease vectors from the genus *Stomoxys* sp. (stable fly) and *Tabanus* sp. (horse fly). These insects mechanically transmit the virus from an infected animal to a healthy animal, during the blood meal of the insect [13]. Tabanidae vectors do not enter the premises where animals are. For that reason, climate conditions, proximity of the forest, shelters, etc., also have an impact on the transmission of the infection.

The acute infection phase lasts from 1 to 3 weeks, but it can be up to 3 months. The EIAV has a specific tropism towards leukocytes and once an animal is infected, chronic viral infection persists throughout the life time of the animal [14]. The virus is found in the blood, so there is also a possibility of the infection being transmitted as an intrauterine infection, or through milk [12]. Clinical signs include fever, malaise, anorexia and petechial bleeding on the mucous membranes. Very often, infected animals exhibit no symptoms at all, but the occurrence of the severe form of anemia can lead to death. Anemia, jaundice, swelling fever and weight loss occur after 30 or more days [10]. The fact that equine infectious anemia virus can reproduce only in equids [11], means that infected horses, donkeys and mules are a constant reservoir of the virus in nature and contribute to the epizootiological features of this disease [15].

Seroprevalence surveys of EIA in donkeys in the world are very limited. Serological investigations have been conducted in Ethiopia, Sudan, Brazil and Italy with the seroprevalence ranging from 8.78% in Sudan to 0% in Italy [10,11].

### 1.2. African Horse Sickness (AHS)

African horse sickness (AHS) is a disease caused by the African horse sickness virus (AHSV), which belongs to the family *Reoviridae*, genus *Orbivirus*. African horse sickness is a highly pathogenic, arthropod-borne viral infection, since it is transmitted by midges belonging to *Culicoides* species. So far, it has been established that the AHSV is transmitted by at least two types of midges and that there are nine different serotypes of AHSV. The disease also has a seasonal character, so it most commonly occurs during the wet periods of the year. African horse sickness can affect all equids (horses, donkeys, mules and zebras). Clinical cases in dogs have also been reported [16].

The first major outbreak of AHS in horses broke out in 1719 in Africa, when 1,700 horses died [17]. Over time, African horse sickness has become a widespread infection in sub-Saharan Africa and nowadays is endemic in that region. It is known that African donkeys and zebras are resistant to infection or the disease has an asymptomatic character, but they serve as a reservoir of the AHS virus. The mortality rate in horses is 70–95%, mules around 50%, and donkeys around 10% [18].

The animal becomes infected through a bite of an infected vector. The clinical manifestations of the disease include a pulmonary form, cardiac form, mixed form and in the form of horse sickness fever. The most common is the mixed form, which is a combination of the pulmonary and cardiac forms, and occurs as an acute infection where the mortality rate can reach 70%. This form is characterized by edema, fever, petechial bleeding, severe dyspnea, cough and colic. Death occurs in 3 to 5 days [19].

African horse sickness is currently limited to the Sub-Saharan region where it is responsible for deaths of horses every year [20]. However, several enzootics in the 1959–1965 period were also recorded outside of the African region, in Asia and in North Africa. After this, the disease moved to Europe, specifically, to Spain. In 1988 the disease broke out again in Spain, killing almost 13,000 horses [21]. During 2020, Thailand faced its first outbreak of AHS, with over 500 dead horses [22].

Epidemiological studies related to the seroprevalence of AHS in donkeys are limited mainly to African region. According to the available literature data, the established seroprevalence of AHS in donkeys in Ethiopia was 32.5%, 75% in Zimbabwe, from 27.6% to 35.2% in Kenya, 63.5% in Namibia, 73% in Uganda and 82.33% in Cameroon [23,24]. Serbia is considered free from AHS.

### 1.3. Equine Herpesvirus Infection (Equine Rhinopneumonitis)

Equine herpesviruses—EHVs represent a group of pathogens that differ from each other based on biological and antigenic characteristics. So far, a total of 9 equine herpesviruses have been identified, six of which belong to the *Alphaherpesvirinae* subfamily and three to the *Gammaherpsevirinae* subfamily of the *Herpesviridae* family. Within both subfamilies, there is a total of six Asinine herpesviruses (AHV-1 to AHV-6) that are related only to the causative agents of the disease in donkeys [25,26].

The most important and clinically significant are equine herpesvirus type-1 (EHV-1) and equine herpesvirus type-4 (EHV-4) (or equid alphaherpesvirus 1 and 4, by the new nomenclature) that belong to the *Alphaherpesvirinae* subfamily, *Varicellovirus* genus. Both viruses are ubiquitous in most equids worldwide and cause a disease called equine herpesvirus infection (equine rhinopneumonitis) and viral abortion, which can also affect donkeys [27].

The upper parts of the respiratory tract are the “entrance door” of infection where EHV-1 primarily multiplies and then penetrates into the circulation via infected leukocytes causing viremia. Equine herpesvirus type-1 further spreads through the blood to the central nervous system (CNS) and other organs (pregnant uterus, eye, etc.) [28,29,30]. On the contrary, EHV-4 virus replication remains restricted to the upper respiratory tract [31].

The intensity of the symptoms of the disease can vary from the subclinical form to the appearance of the clinical signs with respiratory problems, abortions and neurological manifestations leading to a possible fatal outcome. Herpesvirus-1 causes a more serious clinical picture, while the infection caused by EHV-4 is mainly characterized by milder respiratory symptoms. One of the most important characteristics of EHV-1 is the ability to establish a latent infection. Latently infected horses with EHV-1 are permanent carriers of the virus and therefore represent a potential risk for other horses. Reactivation of the virus is possible at any time, and numerous factors can lead to it, such as stress, vaccination, surgical interventions, transport, pregnancy, lactation, etc. [28,32].

Research results related to the seroprevalence of EHV-1 in donkeys vary greatly from country to country and region to region. In two independent investigations in Turkey, antibodies against EHV-1 were detected in 24.2% and 51.85% donkeys, respectively [33,34]. In Ethiopia, the determined EHV-1 seroprevalence in two studies was 20.2%, i.e., 74.7% [35,36]. In addition, in two studies conducted in Egypt, the established seroprevalence was 37% and 71%, respectively [37,38]. Anti-EHV-1 antibodies were detected in 69.5% of tested donkeys in Sudan and in 47% of donkeys in Brazil [39,40]. Additionally, the detection rate of EHV-1 in wild donkeys kept in national parks, reserves or zoos have also been published [41,42,43,44].

### 1.4. Equine Influenza (EI)

The causative agent of equine influenza (EI) is the equine influenza virus (EIV) from the family *Ortomyxoviridae*, genus *Influenzavirus type A*. Influenza viruses are subject to constant changes that can occur in two ways, as antigenic drift and antigenic shift. Antigenic drift represents minor changes in the form of gene mutations. Antigenic shifts are major changes resulting in the reassortment of two different influenza viruses that can produce a novel influenza virus highly contagious for humans [45]. However, in equids, the two most significant subtypes are H7N7 and H3N8. Subtype H7N7 was isolated in 1956, during an epizootic of horse influenza in Czechoslovakia [46] equids. The second subtype, H3N8, was isolated in 1963 in Florida (Miami) [47]. This serotype is widespread in the world posing a great health threat to all types of equids. Donkeys are also susceptible to influenza virus and can become ill in the same way as horses.

The significant risk for an outbreak of EI represents the uncontrolled movement and migration of donkeys [48]. The spread of EIV, in addition to direct contact, is also possible through the infected equipment, vehicles and clothes of the people who work with the animals because the virus can remain infectious for several days [49]. However, the most common way of EIV spreading is through aerosols by the sneezing and coughing of an infected animal [50].

The influenza viruses have the ability to be transmitted from one species of animal to another, including humans, which further complicates the epizootiological surveillance of equine influenza and should not be neglected. For example, EIV subtype H3N8, as the dominant causative agent of equine influenza, can appear in dogs and birds, so these species can also contribute to the occurrence of the disease. In the USA, the occurrence of canine influenza with subtype H3N8 was also recorded [51]. In China, a large number of horses fell ill with influenza, and the disease was caused by the bird influenza virus H3N8 [52]. It is considered that EIV cannot be transmitted to humans, which has not been reliably proven, and the possibility that this disease does not have zoonotic potential cannot be ruled out [53].

According to the available literature, studies of seroprevalence in donkeys are limited. An epidemic caused by EIV (subtype H3N8) was recorded in the entire territory of Chile in horses, donkeys and mules [54]. More extensive research on the large donkey farms in China reported an EI seroprevalence of 32.5% [55]. Equine influenza in donkeys often occurs in the countries of the African continent. In 2019, a large-scale outbreak of EI among donkeys was registered in Senegal [48].

### 1.5. Equine Viral Arteritis (EVA)

Equine viral arteritis (EVA) is caused by equine arteritis virus (EAV). The virus was isolated in 1953 in the USA [56] and since then, the disease has been registered in several countries, on almost all continents. Equine arteritis virus was isolated from the lungs of an aborted fetus, after abortions in stables and a mass outbreak of respiratory syndrome. The virus is classified in the *Arteriviridae* family and *Arterivirus* genus [57].

Equine arteritis virus multiplies in macrophages, endothelial cells and the muscular part of small blood vessels, primarily arteries, with the consequent development of various degenerative and inflammatory processes. In addition, one of the characteristics of EAV is that it can cause asymptomatic (persistent) forms of infection in its host. However, under certain conditions, this virus can lead to the development of severe symptoms of the disease, such as massive bleeding and exudation, even with a fatal outcome, which appears due to the occurrence of superinfection, especially with bacteria, or reactivation of latent infection, such as herpesvirus infections. So far, only one subtype of EAV has been identified. However, there are literature data that indicate a high diversity of isolated viruses [58].

There are limited data in the literature on the prevalence of EAV in the population of donkeys and mules. Thus, in Turkey, Yildirim et al. [59] examined 76 donkeys and reported a seroprevalence of 14.47%. However, in more extensive research in 6 provinces of Turkey, a total of 1532 donkeys were tested and the established overall seroprevalence was 3.45% [60]. Researchers from Bulgaria examined a total of 192 donkey blood sera from 3 regions of Bulgaria and reported a very high overall seroprevalence of 79.1% [61].

## 2. Materials and Methods

The study was conducted by examining 53 out of 199 donkeys that are bred in “Zasavica”. The animals were randomly selected. The age of animals was three to ten years, except for the eight-year-old category, without a single animal. The largest number of animals, 45 (89.4%), was aged from 4 to 6 years. All animals were healthy, without any clinical symptoms of the disease and were not vaccinated against any of the above-mentioned diseases. Figure 2 shows the age distribution by years of the analyzed donkeys.

Blood sampling and tests were conducted during July and August 2022. It was conducted by puncture of the v. jugularis and was carried out according to the Rulebook on Establishing the Program of Animal Health Protection Measures of the Republic of Serbia for 2022 [62] based on the mandatory annual monitoring of infectious diseases. Blood was collected in sterile tubes without anticoagulants and immediately after sampling was adequately transported to the laboratories of the Department for Virology and Department for Serology, Immunology and Biochemistry at the Scientific Veterinary Institute “Novi Sad”. Upon reception, the samples were kept at room temperature, in order to separate the blood sera by spontaneous coagulation. Sera were then collected and stored at a refrigerator temperature for up to 48 h and then tested or frozen at −20 °C until analysis.

Examination of specific antibodies against EIAV was performed by AGID. The test was carried out on fresh samples, immediately after blood sampling, according to the recommendation of the WOAH Manual of Diagnostic Test and Vaccines for Terrestrial Animals [63], using an accredited method, and according to the instructions of the kit’s manufacturer. To perform the test, a commercial substrate for the diagnosis of EIAV (gel) was used, together with a commercial enzyme-linked immunosorbent assay (ELISA) “Equine Infectious Anaemia Virus Antibody Test Kit”, manufactured by VMRD, USA. The diagnostic kit contained a positive control sera and an EIAV antigen. In the gel that was poured into the Petri plates, a mold with holes of a certain volume was pressed into, where the examined sera/samples, positive control/positive serum and antigen were dispensed using a pipette. By diffusing through the antigen, the sera and the antigen reacted. The reaction between the positive serum and the antigen, and a potential reaction of the examined serum (if positive) and the antigen, resulted in the formation of a white precipitation line, which helped in the reading of the results. A negative serum will not form a precipitation line with the antigen. Plates were read after 24 h and again after 48 h. 

The ELISA “Ingezim AHSV Compac Plus” kit, produced by Ingenasa, Spain, was used for the detection of antibodies against AHSV, according to the manufacturer’s instructions and in line with the prescribed method of the WOAH Manual of Diagnostic Test and Vaccines for Terrestrial Animals [16]. Positive and negative controls, provided by the ELISA kit’s manufacturer, were included in the test. The O.D. values of positive and negative control serums were 0.089 and 1.542, respectively. In short, the test was based on the principle of the blocking ELISA method. The VP7 recombinant protein of AHSV (serotype 4) was used as the plate-bound antigen. The calculation of the presence of antibodies was based on the blocking percentage (BP). The samples whose BP was lower than 45% were considered negative, more than 50% positive, and between 45 and 50% were suspicious.

The testing for the presence of antibodies against EHV-1 was performed by virus neutralization test (VNT), according to the method described in the WOAH Manual of Diagnostic Test and Vaccines for Terrestrial Animals [30]. In this test, together with the test samples, the RK-13 cell line was used (ATCC: CCL-37, LGC Standards), positive control sera with antibody titer against EHV-1 1:16 (G 157, American Bio Research), negative control sera with antibody titer against EHV-1 < 1:2, as well as the reference virus strain EHV-1 Kentucky (American Bio Research), in a working titer of 133 TCID50/0.1 mL. Before testing, the samples were inactivated at a temperature of 56 °C in a water bath for 30 min. Each test sample was set up in duplicate and in two-fold dilution. The results were read on an inverted light microscope by monitoring the appearance of the cytopathogenic effect (CPE) after 72 h. The neutralization titer was defined as the reciprocal value of the highest serum dilution capable of neutralizing 100% of the viral CPE. An antibody titer equal to or greater than 1:2 was considered positive.

The determination of specific antibodies against the EIV (subtype H3N8) was performed by hemagglutination inhibition (HI) test according to the WOAH Manual of Diagnostic Test and Vaccines for Terrestrial Animals [64]. The reference virus EIV H3N8 (Equine influenza virus H3N8, strain Miami/1/63, the National Veterinary Services Laboratories (NVSL), Ames, IA, USA) was used in the test in a concentration of 4 hemagglutination units together with reference positive and negative antisera EI H3N8 (Equine influenza antisera Miami H3N8, NVSL). Additionally, 0.5% guinea pig erythrocyte suspension and saline solution were used in this method. The tested sera were inactivated in a water bath for 30 min at a temperature of 56 °C. The tests were carried out on serial double-diluted test sera. The determined value of the HI titer was estimated on the basis of the reciprocal value of sera dilution that caused inhibition of hemagglutination. A serum dilution of 1:16 and higher was considered as a positive finding of antibodies against EIV [65].

The determination of specific antibodies against EAV was carried out by two methods: VNT and ELISA. The tests were conducted according to the instructions of the WOAH Manual of Diagnostic Test and Vaccines for Terrestrial Animals [66] and the instructions of the manufacturer of the ELISA set kit (“ID Screen Equine Viral Arteritis Indirect”; Innovative Diagnostics—IDvet, France). Positive and negative controls, from the ELISA kit, were included in the test. The O.D. value for the positive control serum was 1.202 and for the negative control serum O.D. value was 0.001. In VNT, cell line RK-13 (ATCC: CCL-37), virus (EAV, strain CVL Bucyrus; 070-EDV, NVSL, USA), positive serum (370-EDV, NVSL, USA) and negative serum against this virus were used. The tested sera were inactivated in a water bath for 30 min at a temperature of 56 °C. The tests were carried out on serial double-diluted test sera. Sera dilutions of 1:4 and higher were considered positive. The commercial ELISA kit used in the test is based on the principle of the indirect ELISA technique.

Microsoft Excel 365 and ESRI ArcGIS Pro version 3.03 were used for statistical processing and data display. Differences in the seroprevalence between the various categories of donkeys were analyzed using a Chi-square test. The *p*-value < 0.05 was considered statistically significant.

## 3. Results

In blood sera samples of the tested donkeys, specific antibodies against EHV-1 and EIV (subtype H3N8) were detected. No samples tested positive for EIAV, AHSV or EAV.

Antibodies against EHV-1 were registered in all examined blood sera samples of donkeys, with level range from 1:2 to 1:128 (Table 1). A borderline positive finding of the titer value is 1:2 and higher. Most seropositive donkeys, 16 of them (30.18%), had an antibody titer of 1:16, and one animal (1.89%) from the category of 10- and one from 5-years old had an antibody titer of 1:2 and 1:128, respectively. The difference in seroprevalence between these categories was significant (Chi-square test: 15.7634; *p*-value: 0.000072).

According to the presented results of serological tests for equine influenza, an antibody titer of 1:16 and higher was found in 18 samples (Table 2). Therefore, 18 (33.96%) out of 53 examined samples were found to be positive. Positive results were found in the animals of all age categories, except for the 7 years old. The largest number of seropositive animals was found in the category of 5 years old animals—eighteen of them (33.96%) out of the total number of the examined animals.

Figure 3 shows a comparison of the presence of antibodies against EHV-1 and EIV in donkeys in “Zasavica” based on the age of the animals. Most of seropositive animals were 4 to 6 years old, where as many as 45 (84.90%) animals had antibodies against EHV-1, and 14 (26.41%) against EIV. The difference in seropositivity between the animals infected with EHV-1 and EIV were significant (Chi-square test: 36.7349; *p*-value: 0.00001).

## 4. Discussion

The findings obtained after analysis of the presence of specific antibodies against EIAV indicate that this infection is not found in the population of donkeys in “Zasavica”. Based on the research by other authors, such as Cook et al. [67], it seems that although the AGID test is a highly specific test for the identification of infected animals (primarily horses), it might have a lower diagnostic sensitivity in donkeys, due to the small amount of virus present. There is also the later humoral response in donkeys compared to horses [67]. Research conducted in Bulgaria [61] and Turkey [33] resulted also in negative findings for this disease in donkeys. In the region where “Zasavica” is located, an analysis was completed of the presence of specific antibodies against EIAV in horses during a 20 year period (1994–2013), when a total of 11,972 horses were examined, and only 0.17% of animals were positive [68]. In the second study during the 2013–2014 period, in the same geographical region as in the first study, 316 horse blood serum samples were examined and a prevalence of 1.6% was found [69,70].

The results of this research showed that AHS is not found in donkeys in Serbia, in “Zasavica”. This is in line with the research conducted in Bulgaria, where antibodies against AHSV were also not detected in a single donkey out of 192 animals tested [61]. However, some researchers emphasize the fact that there is still a potential threat of African horse sickness in Europe. Namely, AHSV and bluetongue virus belong to the same *Orbivirus* genus and have common vectors—flies from the *Culicoides* genus. During the last decade, due to global warming and increased human and animal migrations, these vectors have expanded their ecological niches [71]. As a result, there was an outbreak of Bluetongue disease in 2006 in Northern Europe, where this disease had never been registered before. Regular monitoring is the only way to prevent the spread of AHS into territories where the disease had not been previously registered [72]. Studies on the seroprevalence of EHV-1 in donkeys vary widely from country to country and region to region, so the level of antibodies detected ranges from 3% to 100%. In “Zasavica”, antibodies against EHV-1 were found in all donkeys, the seroprevalence was 100%. Researchers from Bulgaria, Chenchev et al. [61], found the antibodies against EHV-1 in all tested donkeys, although the average number of seropositive animals was 69.7% (63–100%). In one reserve in Kenya, EHV-1 seroprevalence was 100% in wild donkeys [42], which is also in line with our results. There are several factors that have influenced the significant differences in seroprevalence, primarily related to the type of laboratory technique, the number of tested samples, the season when the sampling was carried out, as well as farm management. In the obtained results of this study, the value of the antibody titer against EHV-1 ranged from 1:2 to 1:128. Serological analysis was performed with VNT also in Egypt, when the highest value of antibody titers against EHV-1 was 1:128 [38].

In this study, EHV-1 seropositivity increased with the age of the animals, so at the age of 3, only 4 (7.55%) donkeys were seropositive, while 45 (84.90%) animals at the age of 4–6 years had antibodies against EHV-1. This finding was expected since older animals were exposed to the virus circulation for a longer time. Similar conclusions were drawn by Mekonnen et al. [36], who found a higher seroprevalence in older donkeys and a lower seroprevalence in donkeys younger than three years (94.7% vs. 63%). It can be observed that the seroprevalence in this study decreases in donkeys older than 6 years. However, the study included a small number of animals aged 7, 9 and 10 (one donkey each aged 7 and 9 and two animals aged 10), which is not a representative number of samples and certainly affects the final outcome and trend of seropositivity.

In “Zasavica”, the simultaneous presence of antibodies against two viruses, EHV-1 and EIV, was found in 33.96% (18) donkeys, indicating the possibility of co-infection. We assume that both viruses have been circulating in the donkey population for a long time. In Bulgaria, Chenchev et al. [61] described the co-infection with EHV-1 and EIV in 12% of examined donkeys. Although the donkeys in “Zasavica” did not show any clinical signs of the disease, some authors have reported that the presence of co-infection with EHV-1 and EIV could potentially increase the risk of viral reactivation in cases of stress due to the movement or mixing of the animals [41,42]. Donkeys and horses in “Zasavica” are kept together, and that cohabitation certainly contributed to the emergence of latent EHV-1 infection in donkeys. Namely, although the horses in the reserve were not tested for the presence of antibodies against EHV-1, several previous studies have shown that EHV-1 has been circulating among horses in Serbia for a long time [73]. One of the first serological studies on EHV-1 in the blood serum of sport horses in Vojvodina was published by researchers Lazić et al. [74], when the neutralizing antibodies against EHV-1 were found in 91 out of 102 examined horses from stables in Vojvodina (90.19%, 90/102), while the seroprevalence in horses from private breeders was 80.34% (94/117). Furthermore, in the 1995–1999 period, 1741 horses were examined, and the largest number of them had an antibody titer value of 1:32 (23%, 401/1741) and 1:64 (451, 451/1741) [75]. In a study in Kenya [43], serological analysis confirmed the presence of anti-EHV-1 antibodies in zebras (84%) and wild donkeys (100%) that were in close contact and bred together, which leads to the conclusion that the infection in donkeys in “Zasavica” probably originates from infected horses.

The determined value of the antibody titer of 1:16 against EIV (subtype H3N8), which is considered a positive finding, was detected in 19 (35.85%) out of the 53 tested animals. The donkeys are bred together in a pasture with 206 horses. We can hypothesize, especially since the clinical manifestation of the disease has not been established in the previous period, that the determined antibody titer is the result of an old infection and EIV circulation in the population of equids in “Zasavica”. However, a couple of serious outbreaks of EI in donkeys are recorded in different regions in the world [48,54,55].

Based on serological findings on EAV, it can be concluded that infection with this virus is not present in the donkey population in “Zasavica”. Although in Turkey and Bulgaria EVA was detected in different locations and in various number of donkeys [59,60,61], the results of this study revealed the absence of antibodies against EAV in the population of donkeys in “Zasavica”.

## 5. Conclusions

Donkeys are considered to be susceptible to the same viral diseases as horses. The objective of this study was to randomly sample 53 of the 199 donkeys housed in “Zasavica” to determine if donkeys had been exposed to five viruses: equine infectious anemia virus, African horse sickness virus, equine herpesvirus type 1, equine influenza virus (subtype H3N8) and equine arteritis virus. Antibodies against EHV-1 and EIV (subtype H3N8) were detected, while no donkeys tested positive for EIAV, AHSV or EAV. For EHV-1, present in 53 donkeys, the antibody titer ranged from 1:2 to 1:128. For EIV (subtype H3N8), present in 18 donkeys, the highest antibody titer was 1:16.

It can be concluded that it is necessary to introduce regular monitoring of donkeys, improve health care and animal welfare in donkey farms or households that keep donkeys. The number of donkeys in Serbia is less then horses, but there is an economic potential for their breeding, especially for milk production purposes. Therefore, it is essential to continue the research, which will provide the basis for creating a program of health control in donkeys and monitoring of infectious diseases.

## Figures and Tables

**Figure 1 animals-13-02056-f001:**
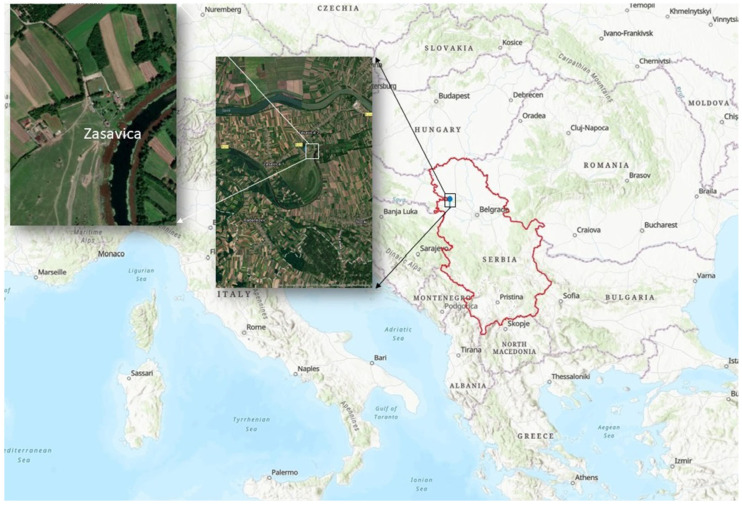
Location of the Special Nature Reserve “Zasavica”, Republic of Serbia.

**Figure 2 animals-13-02056-f002:**
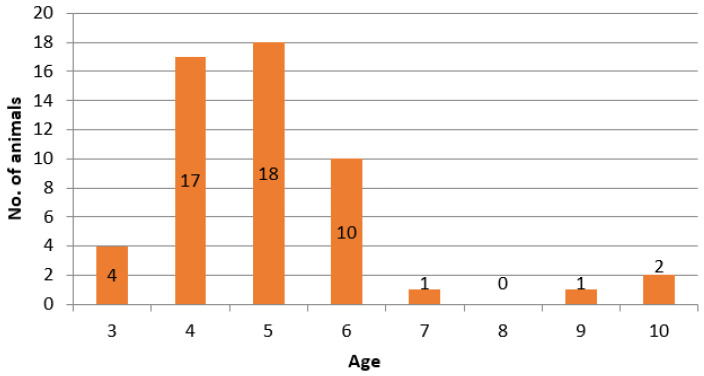
Age categories of 53 randomly selected donkeys in the Special Nature Reserve “Zasavica”, Republic of Serbia.

**Figure 3 animals-13-02056-f003:**
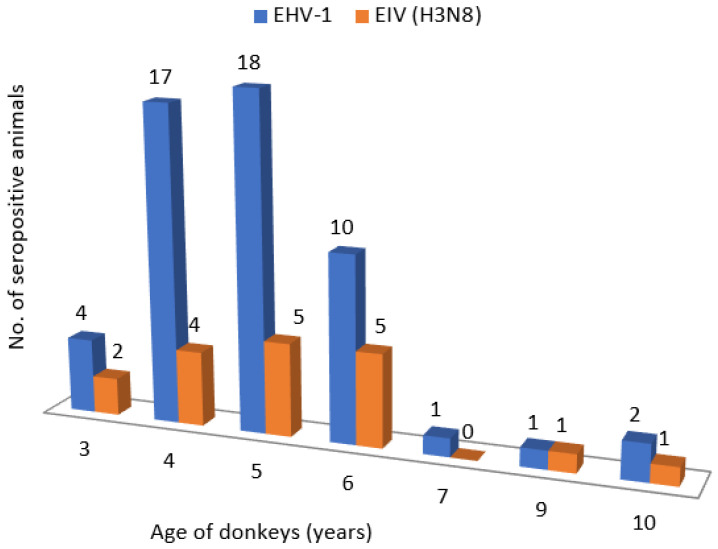
Comparative presentation of antibodies against EHV-1 and EIV in donkeys in relation to the age of the animals.

**Table 1 animals-13-02056-t001:** Distribution of antibody titer against EHV-1.

Donkey Age	Number of Animals with Antibody Titer against EHV-1								Total Number of Animals (%)
	<1:2	1:2	1:4	1:8	1:16	1:32	1:64	1:128	
3	0	0	0	0	3	1	0	0	4 (7.55)
4	0	0	1	1	8	5	2	0	17 (32.07)
5	0	0	3	3	4	2	5	1	18 (33.96)
6	0	0	0	0	1	5	4	0	10 (18.87)
7	0	0	0	1	0	0	0	0	1 (1.89)
9	0	0	0	1	0	0	0	0	1 (1.89)
10	0	1	0	0	0	0	1	0	2 (3.77)
Total seropositive (%)	0 (0%)	1 (1.89%)	4 (7.55%)	6 (11.32%)	16 (30.18%)	13 (24.53%)	12 (22.64%)	1 (1.89%)	53 (100.00%)

**Table 2 animals-13-02056-t002:** Distribution of antibody titer against EIV H3N8.

Donkey Age	Number of Animals with Antibody Titer against EIV H3N8					Total Number of Animals (%)
	<1:2	1:2	1:4	1:8	1:16	
3	0	1	1	0	2	4 (7.55)
4	2	2	0	9	4	17 (32.07)
5	7	3	1	2	5	18 (33.96)
6	4	1	0	0	5	10 (18.87)
7	0	0	0	1	0	1 (1.89)
9	0	0	0	0	1	1 (1.89)
10	0	0	0	1	1	2 (3.77)
Total seropositive (%)	13 (24.53%)	7 (13.21%)	2 (3.77%)	13 (24.53%)	18 (33.96%)	53 (100.00%)

## Data Availability

The data presented in this study are available on request from the corresponding author.

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
