# Peer review of "Serological Examinations of Significant Viral Infections in Domestic Donkeys at the Special Nature Reserve “Zasavica”, Serbia"

_animals, 2023, doi:10.3390/ani13132056_

Round 1

Reviewer 1 Report

[Animals] Manuscript ID: animals-2363695

Manuscript titled “Serological Examinations of Significant Viral Infections in Domestic Donkeys at Special Nature Reserve “Zasavica”, Serbia” by Lazic et al.

General Comments.  This manuscript has repetitive phrases and presents data in both tables and in text.  This reviewer thinks that this manuscript is difficult to read and would need an extensive revision for readability.  The writing should be improved by clear word choices, logical flow of sentences, eliminating excess words, and eliminating repetition. Line 343 states “The study was conducted by examining 53 out of 199 blood sera samples of donkeys…”.  Elsewhere it is written that 53 donkeys were sampled, not 199; the methods do not state that blood samples were taken from 199 donkeys.  Throughout the manuscript, suggest to write more precisely when making a distinction between a virus or a disease.  The Introduction Section includes a lengthy description of each of the five viruses. For each virus, the organization of writing varies and the chosen information varies, making it difficult for a reader to follow and to remain engaged.  Suggest the Introduction be shortened and organized.  The Conclusions Section is poorly written and there is a missed opportunity in this section to summarize the study findings, without jargon words or subjective opinion.  Suggest to delete the Conclusions section or re-write, in brief.  In the References Section, the same article by Câmara et alia 2020 is numbered as reference 25 on line 636, and also as reference 57 on line 709.  Suggest revising this manuscript to focus on only serology results and compare and contrast such results with other herds of donkeys in other geographic areas, and not any lengthy introduction or discussion of these five diseases. 

Line Comments.

Line 31 “SNR” abbreviation is not defined

Line 32 “tested.Specific” missing a space

Line 33 “AGID” abbreviation is not defined.  Replace “the AGID method” with “agar-gel immunodiffusion”

Line 33 “ELISA” abbreviation is not defined

Lines 59 through 64.  Though interesting to read about donkey milk, these sentences seem not appropriate for this manuscript

Lines 77-78 “The largest number of donkeys, 199 of them, lives in the Special Nature Reserve "Zasavica" (Figure 1).”

Line 80: “Figure 1: Location of the Special Nature Reserve “Zasavica”.  Suggest adding to this figure title “Serbia” or “Republic of Serbia”

On Line 78, the reserve name is identified in full “the Special Nature Reserve "Zasavica".  After first use, there is not a need to write out in full “the Special Nature Reserve "Zasavica". In sentences that follow after Line 78, the full name of “The Special Nature Reserve "Zasavica" can be shortened to a place descriptor such as “the reserve” or “Zasavica”.  For example, Line 510 reads “In "Zasavica" the donkeys did not show signs of illness,…” Line 82 “The Special Nature Reserve "Zasavica"; Line 86  “The Special Nature Reserve "Zasavica"; Line 94 “… Special Nature Reserve "Zasavica"

Line 86 “The Special Nature Reserve "Zasavica" has gained the international status from several international institutions and is a member of the Federation of Nature and National Parks of Europe and other protected natural resources”

Line 91 “… 199 domestic donkeys (mainly of the Balkan breed)…”.  Suggest moving “mainly of the Balkan breed” to line 77-78 where “199” first appears in the introduction

Lines 93-96 “The main objective of this study is to show the potential exposure of donkeys in Special Nature Reserve "Zasavica" to the most important viral agents and establish the presence of specific antibodies against Equine infectious anaemia, African horse sickness, equine herpesvirus 1, equine influenza type A (subtype H3N8) and Equine viral arteritis.”  “important” is subjective.  Suggest to rewrite this objective using clear wording, perhaps: “The objective of this study is to test a population of donkeys in "Zasavica" to determine if they have been exposed to five viruses:  Equine infectious anaemia, African horse sickness, equine herpesvirus 1, equine influenza type A (subtype H3N8) and Equine viral arteritis.” 

Line 102-103 “Equine infectious anaemia (EIA) is considered as an extremely contagious disease, which occurs in several ungulate species (horses, mules, donkeys and zebras).”  Suggest instead “Equine infectious anaemia (EIA) occurs in ungulates, such as horses, mules, donkeys and zebras, and is extremely contagious”

Line 103-105 “Equine infectious anaemia is an infectious multisystemic disease, which is listed as one of the 15 notifiable diseases of equides by the World Organization for Animal Health (WOAH) [9].”  This sentence is too complex in that it contains two different points.  Suggest rewrite as two sentences rather than one sentence

Lines 113-114 “The causative agent of equine infectious anaemia can only survive in the body of equids.” Suggest instead “Equine infectious anaemia can survive in the body of only equids [insert reference]”.  This statement is not consistent with the previous statement that the virus can survive on instruments at room temperature

Lines 114-115 “Although this disease represents a health and economic problem in ungulates, the occurrence of EIA in donkeys has not been extensively researched and is often a neglected area.”

Line 119 “Infectious anaemia of ungulates occurs all over the world” Repetitious.  Delete

Line 123 repetitious phrase “Infectious anaemia of ungulates”

Line 132-133 “…also affect the transmission of the infection”.  Suggest changing the word “infection” to “pathogen” or “virus”

Line 160-161 “It has also been confirmed that dogs can acquire the disease with the severe clinical signs [16].”

Lines 301-303 “Therefore, these findings on seroprevalence in donkeys in the Special Nature Reserve "Zasavica" are the first results of these serological tests in the Republic of Serbia.”  Delete sentence, already stated elsewhere in manuscript

Line 343 “The study was conducted by examining 53 out of 199 blood sera samples of donkeys…”.  According to the Methods, blood was drawn from 53 donkeys and not drawn from all 199 donkeys

Line 352 “Figure 2: Number of examined donkeys by age”.  Suggest “Figure 2: Age categories of 53 randomly selected donkeys in the Special Nature Reserve Zasavica, Republic of Serbia

Line 354-356: “Blood sampling was conducted by puncture of the v. jugularis and was carried out according to according to the Rulebook of the Program of Measures for Health Protection of Animals in the Republic of Serbia for 2022 [61] based on the mandatory annual monitoring of infectious diseases.

Line 357-358: “Blood was collected in sterile tubes without anticoagulants and immediately after sampling was adequately transported to the lab.”

Line 358 “… transported to the lab.”  Please insert the name and location of the lab

Lines 361-362 “Blood sampling and tests were conducted during July and August 2022.” Suggest moving this sentence to line 354 to be the first sentence of this paragraph

Lines 366: “… WOAH Manual of Diagnostic Test and Vaccines for Terrestrial animals…”  A reference is expected following the first mention of this manual but is absent at Line 366.  In comparison, see Line 381-382 where there is a reference “Manual of Diagnostic Test and Vaccines for Terrestrial animals [16]”.  Regarding the name of the Manual, seen in this manuscript are variations of spelling: “Test” “Tests” “animals” “Animals”.  See lines 388-389 “"Manual of Diagnostic Tests and Vaccines for Terrestrial Animals [62]”.  At Line 388-389, the name of the manual is not the same.  At line 381-382 and at line 388-389, the reference is not the same as there is a “16” and a “62”.  At Line 401-402 “WOAH Manual of Diagnostic Test and Vaccines for Terrestrial animals [63]” At Line 647 is the name “Manual of Diagnostic Tests and Vaccines for Terrestrial Animals” “Animals” instead of “animals”

 Lines 366-367: “… WOAH Manual of Diagnostic Test and Vaccines for Terrestrial animals, using an accredited method, and according to the instructions of the kit manufacturer.”  please insert the name of the kit.  If the name of the kit is as written on Line 370, please eliminate the excessive wording that exists between Lines 366-371

Lines 367-368 “The tests were carried out at the Scientific Veterinary Institute "Novi Sad".”  Is this the same lab location as line 358?

Line 378: “Plates are read after 24 and 48 hours.”  Are plates read two times?  If yes, please adjust the wording to be clear.  Suggest “Plates are read after 24 hours and again after 48 hours.” 

Line 379 “ELISA” abbreviation is not defined. 

Line 379 has no location stated for “produced by Ingenasa” whereas Line 370-371 states a location “manufactured by VMRD, USA”.  In the Methods, for each text kit used, insert the name of the manufacturer and the location of the manufacturer

Line 388 “… by a virus-neutralization test (VNT),…”.  Here, VNT abbreviation is defined.  In the remainder of the manuscript, “VNT” does not appear.  In the remainder of the manuscript is “virus-neutralization test” and “virus neutralization test”

Line411-412 “Determination of specific antibodies against EAV was carried out by two methods: virus neutralization test and ELISA”.  At line 387-388 it states “virus-neutralization test (VNT)”

Line 427 “In blood sera samples of the tested donkeys, specific antibodies against EIV H3N8 and EHV-1 were detected, while no samples tested positive for EAV, EIA and AHS.”  Please re-order this list according to the order presented in the Methods:  EHV-1 (line 387) then EIV H3N8 (line 400) then EIA (line 363) then AHS (line 380) then EAV (line 411).  Suggest to instead write “In blood sera samples of the tested donkeys, specific antibodies against EHV-1 and EIV H3N8 were detected. No samples tested positive for EIA, AHS or EAV.”

Line 435 Table 1 and line 441 Table 2.  Please reverse the order.  Table 1 becomes “Distribution of antibody titre against EHV-1”.  Table 2 becomes “Distribution of antibody titre against EIV H3N8”

Line 442 the figure legend has repetitious wording of results that can be seen by the reader’s inspection of the 2 tables and by inspection of Figure 1.  Suggest deleting Lines 445-449 “Individually observed, it can be concluded that the most seropositive animals were at the age of 5 years old animals (33.96 %) for the presence of anti-EHV-1 antibodies and 5 (9.43 %) animals had antibodies against EIV. In the 7-year-old category, only one animal (1.89%) had antibodies against 448 EHV- 1, while anti-EIV antibodies were not found

Line 443-444 “The most of seropositive animals…”.  Delete “The”

Lines 461-462: “Considering the lack of research on infectious anemia in donkeys, there is not much information about this disease.”  What’s your definition of “lack of research”. There is information about EIA.  Suggest to delete this sentence or be specific what type of research is lacking

Line 464 “In the region where "Zasavica" is located (southern Bačka and Srem),…”.  Consider whether it is helpful to a reader to include these region names in the Introduction near lines 82-84

Lines 476-477: “During the last decade, due to global warming and increased human and animal migrations, these vectors have expanded their ecological niches of action.”  Please add a reference for this statement.  Suggest to delete “of action”

Line 481 insert space “…tered[70,71]”

Line 481 “Studies dealing with the seroprevalence of EHV-1 …”.  Suggest “Studies of the seroprevalence of EHV-1 …”

Line 540-559.  Conclusions.  This section is poorly written, wordy, redundant and has statements that are not supported by published literature.  Suggest to either delete the Conclusions section or rewrite, in brief, a focused summary such as “The objective of this study was to randomly sample 53 of the 199 donkeys housed in "Zasavica" to determine if donkeys had been exposed to five viruses:  Equine infectious anaemia virus, African horse sickness virus, equine herpes virus type 1, equine influenza type A (subtype H3N8) and Equine arteritis virus.  Antibodies against EHV-1 and EIV H3N8 were detected, while no donkeys tested positive for EIA virus, AHSV, or EAV.  For EHV-1, present in 53 donkeys, the antibody titre ranged from 1:2 to 1:128.  For EIV H3N8, present in 18 donkeys, the highest antibody titer was 1:16

Line 542 “However, they are an unfairly neglected population in the category of equids”.  This is a statement of personal opinion, please delete

Redundancy of References:  Reference 25 appears to be the same as Reference 57.

Line 636: Reference number 25. Câmara, R.J.F.; Bueno, B.L.; Resende, C.F.; Balasuriya, U.B.R.; Sakamoto, S.M.; Reis, J.K.P. dos Viral Diseases That Affect Donkeys and Mules. Animals 2020, 10, 2203, doi:10.3390/ani10122203.

Line 709: Reference number 57. Câmara, R.J.F.; Bueno, B.L.; Resende, C.F.; Balasuriya, U.B.R.; Sakamoto, S.M.; Reis, J.K.P. dos Viral Diseases That Affect Donkeys and Mules. Animals 2020, 10, 2203, doi:10.3390/ani10122203

 Lines 721-722 identified as reference #63 does not include the name of the manual and the word “influenza” is not spelled correctly “World organization for animal health (WOAH). Equine Influenza (Infection with Equine Infleunza Virus), Chapter 3.6.7. ...”

See above "Comments and Suggestions for Authors
"

Author Response

Notes to Reviewer 1:

Reviewer 2 Report

The Lazic et al. manuscript presents the serological study related most important equine infectious diseases as the Equine  infectious anemia,  African  horse  sickness,  Equine  herpesvirus  type  1,  Equine  influenza 20(subtype H3N8), and Equine viral arteritis. This research was done on donkeys from a natural reserve in Serbia, “Zasavica”. Here 199 donkeys are free living in contact with 206 horses and other species.

Lazic et al. have studied have done a serological study of 53 donkeys that were analyzed and found that those were free of Equine infectious anemia,  African  horse sickness,  and Equine viral arteritis. In contrast, some of them were positive to Equine influenza (subtype H3N8) and 100% positive for Equine  herpesvirus type  1.

The manuscript is interesting, with a nice introduction to all the diseases studied in the experimental work but with a limited amount of results and the selection of a not well-justified sample to study the prevalence of the disease. For those reasons, I would like to describe some major and minor concerns after reading this interesting work.

Mayor concerns:

As described in line 343, the authors had 199 blood samples, from donkeys that are bred in the Special Nature Reserve “Zasavica”, and selected randomly 53 samples where  51 were females and 2 males, and the ages of the donkeys were from 3 years to 10. I will find it interesting to include the data from the 199 donkeys and a statistical study that shows that the 53 samples selected are a representative population.

It will interesting to include in the article or explain to the editor why almost 200 donkeys were exposed to a blood sampling, and to the stress induced by this puncture, but only 53 samples were analyzed. It will be interesting to justify why the random selection of the samples to be analyzed was done before collecting the samples from all the animals and not previously to this sample collection, avoiding 146 donkeys an unnecessary puncture.

As described in line 93 the objective of this work is to show the potential exposure of donkeys in the Special Nature Reserve “Zasavica” to the most important viral agents. It will be interesting to describe the exposition of those viruses and compare them with the prevalence of those viral agents on horses living in the same reserve. This will enrich the results section and will feed the discussion section of this work.

Minor concerns:

Line 136 – it is described that EIAV has an incubation period, It will be more appropriate to describe this period as the acute infection phase.

Line 138 – It is described that the EIAV virus found in the blood can be transmitted intrauterine and through milk. We know that other retrovirus can be transmitted in utero and also HIV has been found in milk, but EIAV is another virus that does not have the same cycle, and an actual bibliographic reference to support those ideas it will be welcome.

Line 320 – Varuses –change to Viruses

Line 508 – authors claim a correlation of data with other published data. It will be interesting to explore a bit more this correlation and indicate a value of correlation or a figure representing those points of correlation. 

The English quality is good, in the average of scientific articles

Author Response

Review report to Reviewer 2:

Reviewer 3 Report

Manuscript ID: animals-2363695

Review report on the article entitled Serological Examinations of Significant Viral Infections in Domestic Donkeys at Special Nature Reserve “Zasavica”, Serbia

The authors evaluated the potential exposure of donkeys in a Serbian nature reserve to several viruses that infect members of the family Equidae. Various assays were used to detect antigen-specific antibodies against Equine infectious anaemia virus (EIAV), African horse sickness virus (AHSV), Equine herpesvirus 1 (EHV-1), Equine influenza virus (EIV) subtype H3N8 and Equine arteritis virus (EAV) in blood serum of donkeys. Antigen-specific antibodies were detected against EHV-1 and to a lesser extent EIV subtype H3N8. Surveillance studies are important, which will likely become paramount in the future due to various factors that include global warming and subsequent northward expansion of several viral vectors.

There are numerous instances (simple summary, abstract and throughout the article) where the authors wrote the detection of antibodies against the disease as opposed to the virus. Importantly, Equine infectious anaemia (EIA), African horse sickness (AHS) and Equine viral arteritis (EVA) are diseases respectively caused by EIAV, AHSV and EAV. Additionally, ‘Equine influenza (subtype H3N8)’ is a combination of the disease and virus subtype. Replace the diseases with the correct virus names for the above-mentioned cases throughout the article.

Make sure to include the controls used for all viruses in the Materials and Methods section (the AHSV controls are missing).

It is a requirement for all scientific articles to provide details about the statistical analysis used in experiments. It is unacceptable to only mention the programs used. The authors must clearly state which statistical tests, methods, etc as well as what is considered a significant P-level were used in Microsoft Excel 365 and ESRI ArcGIS Pro version 3.03.

Specific comments:

Whole article: The names, spelling and/or abbreviations for some of the diseases, viruses and other biological classifications are incorrect in multiple places throughout the article, including the headings. Acquire literature about taxonomy, classification and nomenclature; use the correct spelling and abbreviations for the names of the viruses, diseases, vectors and other biological classifications. In addition, check which names should or should not be capitalized or italicized and apply changes where applicable.

1. Introduction to end of article: Write the complete word first and then introduce the abbreviation; only use the abbreviation from that point on. Numerous complete words were used after the abbreviations have been introduced. Correct this.

1.1. Equine infectious anaemia (EIA), Lines 133-135: [“Given that Tabanidae vectors do not enter the premises, climate conditions, the proximity of the forest, the shelters, etc., also have an impact on the transmission of the infection.”] - The sentence is difficult to follow due to its structure. Rephrase this sentence.

1.1. Equine infectious anaemia (EIA), Lines 137-138: [“…..infected, it remains infected forever.”] - Use the correct scientific terminology (e.g. persistent or chronic viral infection). Rephrase this sentence.

1.4. Equine influenza (EIV), Lines 258-261: [“Due to the structure of the genome, Influenza viruses are  prone to imprecise divisions and, consequently, frequent mutations, and therefore are  mentioned as unpredictable causes of influenza, i.e. flu, and represent a constant threat to the health of not only ungulates, but also other types of animals, and humans.”] - A) Include references. B) This sentence does not make sense and is mostly incorrect. Rewrite this sentence (There are many review articles about antigenic drift and shift, and error rates associated with RNA-dependent RNA polymerase during influenza virus replication).

2. Materials and Methods, Line 379: [“The ELISA "Ingezim AHSV Comac Plus" kit”] - Include the controls used.

4. Discussion, Lines 508-512: [“Our results correlate with results from Bulgaria, where dual infection with EHV-1 and EIV was found in 43.8% of examined donkeys [60]. In "Zasavica" the donkeys did not show signs of illness, so we can assume that this co-infection on animals can potentially have an immunosuppressive effect and lead to  the onset of disease in case of stress.”] - The ‘assumption that this co-infection can potentially have an immunosuppressive effect and lead to the onset of disease in case of stress’ is very problematic. Firstly, in addition to not testing for the presence of viruses, there are no references and all assumptions, speculations; etc. must be supported by evidence from published scientific studies. Secondly, influenza viruses cause acute infections (the virus is cleared and the host recovers or the host dies). Thirdly, influenza viruses do not suppress the immune response. In contrast, virulent influenza viruses (similar to other pathogenic ssRNA viruses) interfere with the immune response that result in excessive pro-inflammatory responses, which play a major role in immunopathology and a severe disease outcome. Whereas, well regulated immune responses develop and result in the clearance of less pathogenic influenza viruses. In addition, this ‘co-infection on animals can potentially have an immunosuppressive effect and lead to the onset of disease in case of stress’ is contradictory to the statement below (Lines 529-532). Either provide supportive evidence for this claim or remove it from the article.

4. Discussion, Lines 529-532: [“We can hypothesize, especially since the clinical manifestation of the disease has not been established in the previous period, that the determined antibody titer is the result of an old infection and EIV circulation in the population of equids in SNR “Zasavica”.] - This ‘hypothesizes’ that the influenza virus-specific antibodies were detected after the virus was cleared after a previous infection is much more valid than the ‘assumptions’ made in Lines 508-512 because it was based on the fact that this study did not observe a clinical manifestation of the disease. Furthermore, it is well documented that IgG can still be present in serum long after an infection was cleared; include references for this.

Extensive editing of English language required

Round 2

Reviewer 1 Report

Version 2 of "Serological Examinations of Significant Viral Infections in Do-2 mestic Donkeys at Special Nature Reserve “Zasavica”, Serbia" by authors Sava Lazić, Sara Savić , Tamaš Petrović , Gospava Lazić , Marina Žekić , Darko Drobnjak and Diana Lupulović

General Comment.  The addition of Figure 1, geographic orientation of study area, is appreciated.  The authors continue to make grammatical errors when discussing a disease or a virus causing a disease.  Inconsistent writing and typographic errors are present in version 2.  Examples of errors seen in version 2 are shown below.  Correction by the authors of only these examples does not suggest all other sentences are written in clear concise scientific writing.  Suggest the authors carefully read the manuscript and improve the writing.  Please use clear language when writing about a disease and when writing about a viral causative agent of a disease.       

Line 34. “VNT” is defined in the Abstract but is not defined as first appearance in the main text. 

Lines 61-62. “Donkeys are used in tourism and in animal therapy of children with the developmental disabilities [4,5].”

Line 81 “EUROPARK” abbreviation is not defined. 

Lines 84-87. As pointed out in the review of version 1 of this manuscript, the authors, in version 2 of this manuscript, continue to write incorrectly about a disease or about a viral causative agent of a disease; here in lines 84-87, the “five viruses” include names of diseases:  “The objective of this study is to test a population of donkeys in "Zasavica" to determine if they have been exposed to five viruses: Equine infectious anaemia, African horse sickness, equine herpesvirus 1, equine influenza type A (subtype H3N8) and equine viral arteritis.”

 Line 98-99.  As pointed out in the review of version 1 of this manuscript, the authors, in version 2 of this manuscript, continue to write incorrectly about a disease or about a viral causative agent of a disease: “Equine infectious anaemia can survive in the body of only equids [11].”  

Line 261.  “Zasavica” appears inside quotation marks throughout the manuscript but in Figure 2 is shown without quotation marks: “Figure 2: Age categories of 53 randomly selected donkeys in the Special Nature Reserve Zasavica, Republic of Serbia” 

Line 596.  Reference contains two or more typographic errors “Controls” and “ifluenza”:  “45. Centers for Disease Controls and Prevention (CDC): Ifluenza (Flu)https://www.cdc.gov/flu/about/viruses/change.htm (accessed on 31 May 2023).”

 Lines 337-339. Compared with lines 348-349, there is inconsistent writing “Chi square” “Chi-square” “P value” and “p-value”: “Differences in the seroprevalence between the various categories of donkeys were analyzed using a Chi square test. The P value < 0.05 was considered statistically significant.”

 Lines 348-349. “The difference in seroprevalence between these categories was significant (Chi-square:15.7634; p-value: 0.000072).”

End of provided examples of grammatical errors or incorrect scientific language.  Suggest the authors carefully review all text.

Peer reviewer comments for this manuscript revision are shown above.

Reviewer 2 Report

Dear author and co-authors,

Thank you for considering my review of your manuscript. I appreciate the effort you put into addressing the concerns and questions I raised. I have carefully examined your revised manuscript, and I am pleased to note that you have adequately addressed all of my previous points.

Firstly, I am satisfied with the clarification you provided regarding the number of equids studied.  Your explanation have significantly strengthened the validity and reliability of your findings.

Furthermore, I would like to acknowledge the thoroughness of your response to my queries about the reference of the milk transmission for AIE. The revised manuscript now includes the necessary information, which provides a more comprehensive understanding of the study's context and results.

Overall, I am pleased to acknowledge that your revisions have effectively addressed all of my concerns and have substantially improved the manuscript.

Once again, I would like to express my appreciation for your responsiveness to my comments and the meticulous effort you put into revising the manuscript. It has been a pleasure reviewing your work, and I believe it will make a valuable contribution to the scientific community studying equine diseases.

Best regards

Just a recommendation, double check some mistyping errors.
